# Same accuracy, twice as fast: continual learning surpasses retraining from scratch

## Abstract

Continual learning aims to enable models to adapt to new datasets without losing performance on previously learned data, often assuming prior data is no longer available. However, in many practical scenarios, both old and new data are accessible. In such cases, good performance on both datasets is typically achieved by abandoning the model trained on the previous data and re-training a new model from scratch on both datasets. This training from scratch is computationally expensive. In contrast, methods that leverage the previously trained model are worthy of investigation as they could significantly reduce computational costs. Our evaluation framework quantifies the computational savings of such methods while maintaining or exceeding the performance of training from scratch. We identify key optimization aspects – initialization, regularization, data selection, and hyper-parameters – that can each contribute to reducing computational costs. For each aspect, we propose effective first-step methods that already yield substantial computational savings. By combining these methods, we achieve up to 2.7x reductions in computation time across various computer vision tasks, highlighting the potential for further advancements in this area.

## 1 Introduction

In modern deep learning, the available data tends to expand and evolve over time, often resulting in a model already trained on a large dataset ('*old data*') to be further adapted to perform well also on a new, smaller dataset ('*new data*'). This new data typically introduces slightly different distributions, such as additional classes, new domains, or corner cases. A common approach is to retrain the model from scratch using both the old and new datasets, but as model and dataset sizes increase, it becomes computationally expensive. This highlights the need for more efficient methods that can continuously train models without the cost of full retraining.

A large part of continual learning research has tried to approach this problem in a resource-constrained setting, aiming to reach the highest possible performance on both old and new data under some constraints (Verwimp et al., 2024). Such constraints can lead to suboptimal performance, in part due to catastrophic forgetting (French, 1999). In e.g. industry applications, it is often undesirable to sacrifice performance to save computational costs, and retraining from a randomly initialized model ('*from scratch*') is preferred over using older models (Huyen, 2022). This is a wasteful approach; there is a model available that performs well on the old data, so why not use it? In practice, it has been shown to be difficult to continue to train previously trained models, even without storage constraints on past examples (Ash & Adams, 2020).

In this paper, we focus on the cost of training a model on new data, when a model that performs well on the old data is available. The cost of training this old model is treated as a sunk cost (Kahneman & Tversky, 1972), its training happened in the past and the price for this training has already been paid. In contrast, future costs can be reduced or mitigated, and we show that using old models can be an effective way of doing so. Typically, in computational problems there is both a memory and computational aspect, but when models get large, the computational cost tends to outweigh memory costs. For example, the hard disks to store ImageNet21k are about 50 times cheaper than training a large vision transformer on the same data once (see Appendix A.1 for details of this estimate).

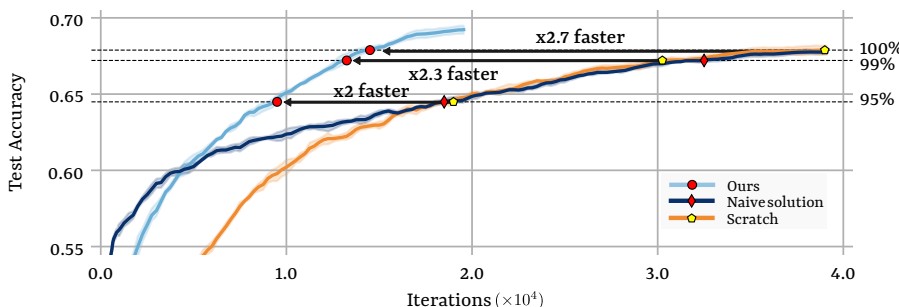

Figure 1: Test accuracy on CIFAR100 (70+30) with a model pre-trained on 70 classes. The 'scratch' method starts from random initialization, while the 'naive' approach uses the pre-trained model without modification. 'Ours' modifies the optimization process (see Section 3) and matches the scratch performance with $2.7\times$ lower computational cost.

While there are good reasons to work in memory-restricted settings (Verwimp et al., 2024), in this paper the focus is on reducing the computational cost when new data becomes available. Particularly, we investigate techniques that can make a difference when an old model is available, instead of general optimization techniques like e.g. mixed precision calculations (Micikevicius et al., 2018).

Instead of the wasteful from-scratch approach, we propose to focus on continuous solutions, which leverage the existing model when new data is available. The simplest of these solutions is to continue training the model on the full dataset, using the previous model's weights for initialization. While this solution uses pre-trained initialization, it converges at a similar speed and often underperforms compared to training from scratch, offering little computational gain. Figure 1 illustrates the limitations of this naive approach.

Here, we introduce an evaluation framework to measure the computational gain of continuous methods. In this framework, we assume access to a previously trained model on the old data, and both the entire new and the old datasets. The goal is to improve the efficiency and accuracy by leveraging this previously trained model, compared to training from scratch on the entire dataset, as well as naive continuous training. We measure the advantage of an approach by the reduction in computational cost required to reach the same accuracy, on the full dataset, as a model trained from scratch.

In Section 3, we start from the canonical SGD update rule and show that all of its aspects – the initialization, the objective function, the sampling process, and the hyper-parameters – can significantly reduce computational costs. We outline and evaluate several strategies – inspired by the literature – that act on these aspects. Each of them presents a promising avenue for future research to accelerate the convergence of continuous models. In Section 4, we show that while these methods individually reduce computational costs, their effects are to some extent complementary; combining them yields even greater reductions. Finally, we further demonstrate that these methods improve training efficiency across a variety of image classification datasets, multi-task settings, and domain incremental scenarios, highlighting the robustness and potential of our method to reduce the computational burden of re-training machine learning models.

## 1.1 OUR CONTRIBUTION

- We propose a novel continual learning evaluation framework, allowing models full access to previous data and measuring their computational cost rather than only their accuracy. This approach shifts the research focus towards more practical, real-world challenges.

- We discuss the different areas in the optimization process that can be explored to accelerate convergence. Each area is broad enough to be the focus of a different method.

- For each area, we introduce a first-step method based on the literature, which already significantly enhances the learning speed of continual models.

- We demonstrate that improvements in these areas are complementary, meaning advancements in one area do not preclude further gains in others.

- Through an empirical study, we show that our combined methods consistently accelerate convergence across multiple datasets and tasks.

## 1.2 RELATED WORK

**Continual learning.** Continual learning concerns cases where data is not available all at once (see e.g. (De Lange et al., 2021; Wang et al., 2024) for surveys). Most studies have focused on settings with strong constraints on the amount of old data that can be stored (Verwimp et al., 2024). Replay methods (Chaudhry et al., 2019; Buzzega et al., 2020) aim to use this stored data as effectively as possible. However, even with replay mechanisms, learning new data often leads to (catastrophic) forgetting of previously learned examples. Other approaches modify the model architectures (e.g. Yan et al., 2021) and regularization losses (e.g. Li & Hoiem, 2017) to reduce forgetting. In contrast, some works have looked at settings where computational cost is restricted rather than memory costs. In these cases, standard replay outperforms other methods (Prabhu et al., 2023). Later works (Smith et al., 2024; Harun et al., 2023) improved replay techniques in these settings. Our work, however, imposes no memory restrictions and instead focuses on accelerating the learning compared to models trained from scratch, a challenge that is in some cases harder than outperforming standard replay on a continuous model, see Section 3.1.

**Faster optimization.** At its core, machine learning involves solving a challenging and computationally expensive optimization problem, and many approaches have been proposed to streamline this process (Sun et al., 2019). First-order methods, such as stochastic gradient descent (SGD)(Robbins & Monro, 1951), are the most widely used, where the full gradient is typically approximated using small batches. However, SGD can suffer from slow convergence, especially in high-variance settings(Johnson & Zhang, 2013), which can be mitigated by techniques like Nesterov momentum (Sutskever et al., 2013). Adaptive approaches such as AdaGrad (Duchi et al., 2011) and Adam (Kingma, 2014) help by adjusting the learning rate dynamically, though explicit learning rate scheduling often enhances their performance (Loshchilov & Hutter, 2022; Smith & Topin, 2019). Second-order methods, despite their promise, are hampered by the computational expense of estimating the Hessian (Martens, 2016). Goyal et al. (2017) also highlights the intricate relationship between batch size and learning rate in neural network optimization. Additionally, regularization techniques like batch normalization (Ioffe & Szegedy, 2015) and weight normalization (Salimans & Kingma, 2016) can further improve convergence. These considerations become especially important when optimizing from a pre-trained model, as is the case in this work, rather than from random initialization (Narkhede et al., 2022).

**Warm start.** Starting from non-random initialization is most commonly used in transfer learning, where a pre-trained model (typically on ImageNet) serves as a starting point to kick-start training downstream tasks (Zhuang et al., 2020). Contrary to our work, these works are often not concerned with performance on the pre-training (i.e. old) data. While beneficial, pre-training may hurt downstream performance (Zoph et al., 2020). This may be explained by a loss of plasticity in trained networks (Dohare et al., 2024; Abbas et al., 2023). Ash & Adams (2020) showed that when continuously training on the same data source, lower performance is reached than when starting from scratch. Gupta et al. (2023); Parmar et al. (2024) study how to continually pre-train NLP models which is strongly related to our setting. They examine learning rate scheduling but do not consider loss of plasticity, regularization, and data selection as done in this paper.

## 2 PROBLEM DESCRIPTION

We consider the following setup: an existing dataset, denoted as $\mathcal{D}_{old} = (\mathcal{X}_{old}, \mathcal{Y}_{old})$, where $\mathcal{X}_{old}$ represents the input examples and $\mathcal{Y}_{old}$ the corresponding labels. A model $f_{old} : \mathcal{X}_{old} \rightarrow \mathcal{Y}_{old}$, has already been trained on this dataset. We are then provided with a new data, $\mathcal{D}_{new} = (\mathcal{X}_{new}, \mathcal{Y}_{new})$, which may include new classes. The objective is to get a model $f : (\mathcal{X}_{old} \cup \mathcal{X}_{new}) \rightarrow (\mathcal{Y}_{old} \cup \mathcal{Y}_{new})$ that performs well on both the new and the old datasets, $\mathcal{D} = \mathcal{D}_{old} \cup \mathcal{D}_{new}$, with minimal computational cost.

To ensure a fair comparison between models, we use the same architecture when comparing the computation costs of different training methods. This, and keeping a fixed batch size, allows us to quantify computational cost through the number of training iterations. Let $f^i$ represent the model $f$ after $i$ training iterations. We define the speed $s$ of a model $f$ in achieving a target accuracy, $a$, as

the first iteration in which $f$ reached or surpassed that accuracy. Formally:

$$s(f,a) = \min_i \left\{ i \in \mathbb{N} : \frac{1}{|\mathcal{D}|} \sum_j^{|\mathcal{D}|} \mathbb{1}[f^i(x_j) = y_j] \geq a \right\} \tag{1}$$

We use the relative speed-up $L_r$ when comparing the performance of different models against a baseline model. The baseline model $f_{scratch}$ is trained from scratch on the combined dataset $\mathcal{D}$ and achieves an accuracy of $a_{scratch}$. Then we can define $L_r$ as:

$$L_r(f, f_{scratch}) = \frac{s(f_{scratch}, a_{scratch})}{s(f, r/100 \cdot a_{scratch})} \tag{2}$$

or the relative number of iterations that a model $f$ requires to reach the same ($L_{100}$) or a fraction of (e.g. $L_{99}$) the final accuracy of a model trained from scratch. For instance, $L_{99} = 2$ would indicate that model $f$ attains an accuracy of $0.99 \cdot a_{scratch}$ with a 2 times lower computational cost than was required to train the full model from scratch to attain $a_{scratch}$, or equivalently, half the number of iterations. To reduce notation complexity, we will simply use $L_r$ in the remainder of the paper when the models can be inferred from context. Note that this measure only works when each iteration has the same computational cost, but the idea can easily be extended to when this is not the case.

## 2.1 IMPLEMENTATION DETAILS

**Datasets.** We conducted experiments on a variety of image classification datasets, including CIFAR-10 (Krizhevsky et al., 2009), CIFAR-100, subsets of ImageNet (ImageNet-100 and ImageNet-200) (Deng et al., 2009), and Adaptiope (Ringwald & Stiefelhagen, 2021). For continuous training, each dataset was divided into disjoint subsets, and training proceeded in a cumulative manner. For example, in CIFAR-100 (80+10+10), classes are split into three groups: 80 classes, followed by 10, and 10. The model was trained sequentially, first on the 80 classes, then on the combined 80 + 10, and finally on all 100 classes. Class splits were randomized, where the specific seeds are available in the attached code.

**Training and baselines.** Unless otherwise specified, we used ResNet-18 with a cosine annealing scheduler (Loshchilov & Hutter, 2016), the Adam optimizer, and a learning rate of 0.001. All models are trained with standard cropping and horizontal flipping augmentations. Additional model and hyperparameter details can be found in the attached code and Appendix A.2. The *scratch* baseline indicates a model that is trained from a random initialization on both old and new data together. The *naive* baseline represents a continuous model that simply continues training from the old model, without any modification.

**Experimental details.** Every experiment shown in this paper is repeated five times. The results shown are the averages of these experiments, accompanied by their standard error in plots. The results presented are always obtained by using the combined test sets of the old and the new datasets.

## 3 METHOD

In deep learning, optimization is typically performed using variants of stochastic gradient descent, where model parameters $\theta_i$ are updated by subtracting an estimate of the gradient of the objective function, based on a small batch of samples. The standard minibatch SGD update rule is:

$$\theta_{i+1} = \theta_i - \frac{\eta}{N} \sum_{j \in B_i} \nabla L(\theta_i, (x_j, y_j)) \tag{3}$$

where $\eta$ is the learning rate, $L$ is the objective function, and $B_i$ a batch of $N$ examples sampled from the full dataset $\mathcal{D}$. All components of this update – model initialization ($\theta_0$), batch composition, learning rate adjustments, and modifications to the objective function – play a critical role in the optimization trajectory and convergence speed. For other optimizers like e.g. Adam (Kingma,

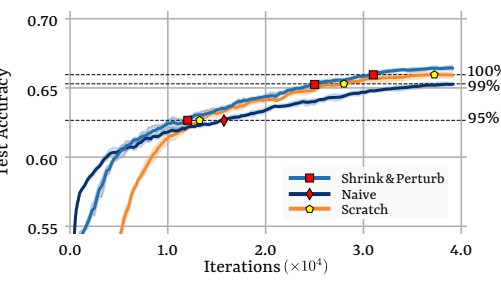 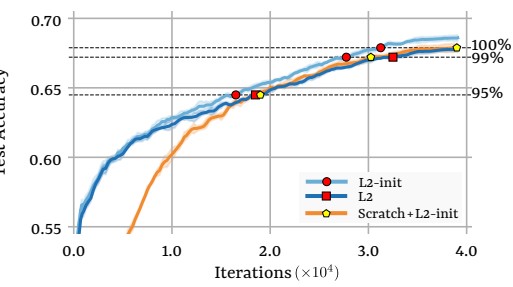

Figure 2: **Initialization**. Naive continuous training is slower and less accurate than retraining from scratch. Re-introducing plasticity with shrink-and-perturb improves both speed and accuracy, surpassing scratch training.

Figure 3: **Objective function**. Regularizing the objective function with $L2$-losses is beneficial in both from scratch and continuous learning, yet the latter outperforms the former when using $L2$-init regularization.

2014), the simple average here would be replaced by a momentum-based average, but the idea remains the same. In the following sections, we explore how each of these elements can significantly accelerate continuous model training. CIFAR-100 (70+30) serves as a case study for comparing various strategies.

### 3.1 INITIALIZATION

When training models from scratch, careful random initialization offers several advantageous properties (Narkhede et al., 2022). However, during continuous training, many of these advantages are lost as training does not start from a random set of weights. Several works have shown that this potentially leads to reduced plasticity, i.e. not being able to learn new information as fast and accurately as a model trained from scratch (Ash & Adams, 2020; Dohare et al., 2023). This issue is similarly observed in continuous training, as shown in Figure 2. The naive benchmark is both worse and converges slower than retraining from scratch.

To restore plasticity during warm-start training without distribution shifts, Ash & Adams (2020) introduced the 'shrink and perturb' method. Rather than using the previously trained model's weights directly, the weights are shrunk by a factor $\alpha$ and combined with a small portion of randomly initialized parameters $\theta_{random}$. The resulting initialization $\theta_{init}$ is computed as:

$$\theta_{init} = \alpha\theta_{old} + \beta\theta_{random} \tag{4}$$

In our experiments, we used $\alpha = 0.4$ and $\beta = 0.001$ without tuning, as proposed by the original authors. However, a broad range of $\alpha$ and $\beta$ values yielded qualitatively similar results (see Appendix A.3). In Figure 2, we compare ResNet-18 models trained continuously on CIFAR-100 (70+30) with and without the shrink-and-perturb method. The results show that re-introducing plasticity can not only accelerate convergence but also potentially improve final accuracy compared to both scratch training and continuous training without it.

### 3.2 OBJECTIVE FUNCTION AND REGULARIZATION

An alternative approach to tackle the reduced plasticity problem is to prevent it from arising in the first place by changing the objective function $L$. Dohare et al. (2023) proposed maintaining plasticity through continual backpropagation, while Kumar et al. (2023) introduced a simplified version that uses an $L2$ regularizer towards the initial random weights $\theta_0$, rather than towards the origin as is typically done in $L2$ regularization. In our setting, this involves modifying the objective function $L$ to include this $L2$-init regularizer:

$$L_{reg}(\theta_i, (x_j, y_j)) = L(\theta_i, (x_j, y_j)) + \lambda\|\theta_i - \theta_0\|^2 \tag{5}$$

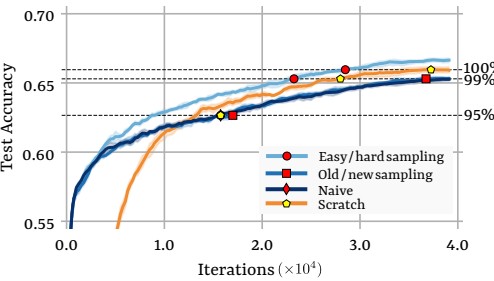 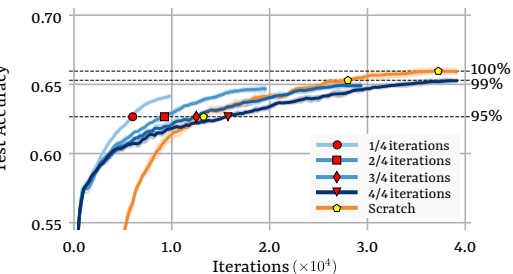

Figure 4: **Batch composition**. 'Old / new' sampling balances old and new examples in each batch, unlike the naive baseline, which uses proportional sampling. 'Easy / hard' sampling reduces the inclusion of the easiest and hardest old examples, significantly improving performance. (Naive and 'old/new' results nearly overlap.)

Figure 5: **Hyperparameters**. Shortening the learning rate scheduler allows for faster convergence but at the cost of lower final accuracy. On its own, changing the scheduler does not reach the required accuracy, yet when combined with the other aspects, it becomes important (see Section 4.1)

In our experiments, we used $\lambda = 0.01$ without tuning, as proposed by the original authors. However, a broad range of $\lambda$ values yielded qualitatively similar results (see Appendix A.4). In Figure 3, we train ResNet-18 models on CIFAR-100 (70+30) and compared the results of models trained from scratch with those trained continuously, both with the $L2$-init regularizer. The results indicate that adding this regularization term accelerates the convergence of the continuously trained models, more so than it improves models trained from scratch. Since regularization can also benefit models trained from scratch, we use this baseline here, as well as the more standard $L2$ regularization (i.e. $\theta_0 = \mathbf{0}$) which is not as effective as $L2$-init regularization.

## 3.3 BATCH COMPOSITION

Estimating the full gradient of a dataset is expensive in the deep learning case, as many iterations are required to reach a good solution. To overcome this, minibatches containing a small part of the entire dataset are used to estimate the gradient. Typically, examples are sampled from the full dataset with uniform probability. In continual learning, it is a common practice to balance examples from old and new data in a batch (Rolnick et al., 2019), which either increases or decreases the sampling probability of old data depending on whether there is either more or less old than new data.

In Figure 4, we show that having an equal number of new and old examples ('old / new sampling') does not improve the results compared to the naive approach, which balances old and new examples in a batch according to their ratio in the full dataset (i.e. 70% old examples and 30% new ones in this example). Katharopoulos & Fleuret (2018) show that the optimal sampling distribution is proportional to the gradient norms of the individual examples. In Appendix A.7, we show that at the very start of training the gradient norms of the old examples are indeed smaller, but after about 100 iterations, there is no noticeable difference on the class level, which explains the results.

While the ratio of old and new examples does not have an immediate effect, not all data in the replay memory is equally useful. Often in continual learning, access to the old dataset $\mathcal{D}_{old}$ is limited to a small fraction. In our case, the replay memory has infinite capacity. We build on the work of Hacohen & Tuytelaars (2024), who propose a sampling strategy that reduces the importance of very easy and very difficult examples in the memory. More formally, they define learning speed $ls$ of a sample $x_j$ as the epoch $e$ in which sample is classified correctly:

$$ls(x_j, y_j) = \frac{1}{E} \sum_{i=1}^{E} \mathbb{1}[f^i(x) = y] \tag{6}$$

with $E$ the total number of epochs and $f^i$ the model at epoch $i$.

The learning speed is used to order the old examples from easy to hard, given that the necessary information is recorded during training of the old model (For more details, see Appendix A.5).

Using this order, we reduce the sampling probability of the 10% easiest and highest examples to one-tenth of the other examples. In the easy examples, there is no information left, and the hardest ones are never learned, and thus as useful. The results in Figure 4 show that this approach ('easy / hard sampling') is helpful and speeds up training in the continuous case. In Appendix A.5 we show that this is robust to the exact hyperparameters and that, while sacrificing some convergence speed, the easiest and hardest examples can be removed entirely.

### 3.4 Learning rate scheduling

The learning rate controls the step size in stochastic gradient descent, directly influencing convergence speed. In deep learning, the learning rate typically starts high to allow faster progress toward optimal solutions and decreases gradually to avoid overshooting (Zeiler, 2012). Common scheduling strategies include 'multistep' scheduling (Zagoruyko & Komodakis, 2016), which reduces the learning rate at set intervals, and cosine annealing (Loshchilov & Hutter, 2017), which decays it more smoothly over time.

When training a model continuously, since the model is already trained on parts of the data, a more aggressive learning rate scheduling can accelerate convergence. While the learning rate still needs to decrease over time, this can happen more rapidly over fewer iterations, as the model has already learned key information. As shown in Figure 15 in the Appendix, the loss curve for the continuous model reaches a plateau much faster compared to scratch training, indicating that the model approaches a local optimum more quickly. This supports the need for a faster reduction in the learning rate (Zeiler, 2012).

We train ResNet-18 models on CIFAR-100 (70+30 split), using different lengths of cosine learning rate schedulers. The most aggressive scheduler used only 25% of the total iterations, while others used 50% and 75%. The learning curves, depicting the mean test errors from these experiments, are shown in Figure 5. The results indicate that more aggressive schedulers lead to faster convergence of the continuous model to a performance level comparable to the scratch solution. However, overly aggressive scheduling hurts the final performance, as the networks fail to achieve 100% of the scratch model's final accuracy. On its own, reducing the learning length is not helpful, yet when combined with the other aspects, it becomes important (See Section 4.1). Similar qualitative results were observed with multistep scheduling, see Appendix A.6. For the remainder of this paper, any adjustments to the learning rate scheduler for specific methods are explicitly mentioned, and a comparison to the unmodified variant is provided.

## 4 Results

In the previous section, we explored several key aspects of SGD optimization, and each can serve as the foundation for methods that reduce the computational cost of continuous training. By leveraging insights from various works in the literature, we proposed a method for each aspect, demonstrating the effectiveness of each method individually.

In this section, we begin by showing that these methods are complementary, with their combination further accelerating convergence beyond what is achieved by applying them separately. We show that although the same techniques can lead to better convergence when training from scratch, the benefit is larger in the continuous setting. We then extend this combined approach to various datasets, scenarios, and data splits, demonstrating its applicability across image classification tasks.

### 4.1 Ablations

In the previous section, we introduced several methods aimed at speeding up the convergence of continuous models, each targeting a different aspect of the optimization process. We now examine whether these improvements are complementary or if the benefits of one method negate the effectiveness of others. Table 1 presents results from training five ResNet-18 models and compares all combinations. While not fully cumulative, each combination offers benefits, either in convergence speed or in final accuracy. Additionally, we applied the same techniques to a model trained from scratch, to rule out the possibility that the techniques are *only* improving overall learning capabilities.

Table 1: Ablation results of the four different aspects studied in Section 3 under '*continuous*' on CIFAR100 (70+30). All speed-ups are relative to the model trained from scratch without any modification, '/' indicates the required accuracy was not reached. Each of the aspects individually establishes a speed-up, and combining them improves the results further, albeit not fully cumulative. On the right hand side of the table, '*scratch*' shows the result of a model trained from scratch using the same techniques. Some of them improve the training speed, but to a lesser extent than in the continuous case.

| Initialization | Regularization | Data | Scheduler | Continuous | | | Scratch | | |
|---|---|---|---|---|---|---|---|---|---|
| | | | | Max Acc | $L_{99}$ | $L_{100}$ | Max Acc | $L_{99}$ | $L_{100}$ |
| | | | | 65.26 | / | / | 65.92 | ×1.33 | ×1.00 |
| ✓ | | | | 66.43 | ×1.49 | ×1.20 | 65.06 | / | / |
| | ✓ | | | 68.60 | ×1.94 | ×1.66 | 67.90 | ×1.69 | ×1.51 |
| | | ✓ | | 66.65 | ×1.60 | ×1.31 | 66.61 | ×1.54 | ×1.31 |
| | | | ×0.25 | 64.15 | / | / | 63.72 | / | / |
| ✓ | ✓ | | | 68.91 | ×2.19 | ×1.82 | 68.01 | ×1.75 | ×1.54 |
| ✓ | | ✓ | | 66.76 | ×2.92 | ×2.53 | 66.61 | ×1.54 | ×1.31 |
| | ✓ | ✓ | | 68.86 | ×1.96 | ×1.69 | 68.30 | ×1.67 | ×1.49 |
| ✓ | ✓ | ✓ | | **69.47** | ×2.19 | ×1.84 | 68.31 | ×1.69 | ×1.52 |
| ✓ | ✓ | ✓ | ×0.25 | 68.26 | **×5.73** | **×5.32** | 63.74 | / | / |

The results show that while some optimization aspects have an effect on the scratch models as well (most notably regularization), their influence is always stronger in the continuous case. This is especially clear when shortening the learning rate scheduler, which allows the continuous model to learn faster, yet greatly reduces the final accuracy of the from-scratch model. These results indicate that the proposed techniques are effective in leveraging the benefits of having a model trained on the old data available. Future research targeting different aspects of the continuous learning process could yield further gains in training speed, as multiple optimization strategies can be effectively integrated to speed up the learning.

### 4.2 MULTIPLE TASKS

We used CIFAR100 (70+30) as a case study in Section 3, adding 30 new classes. In many continual learning scenarios, new data is not only added once, but repeatedly. In Figure 6 we show that applying the same aspects on consecutive tasks continues to work as in the one-task case. For each task, we also train a model from scratch on all classes available up to this point. In Figure 6, the yellow dots indicate when the continuous model reaches the same performance ($L_{100}$) as the from-scratch model for that task. For every task $L_{100} > 1$ and gets progressively larger, indicating that the continuous model benefits more than it has trained for longer in the past.

These results highlight an important insight: in total, the continuous models have trained for more iterations than the models that are trained from scratch, which also explains why their final accuracy may surpass that of training from scratch. However, when accumulating past costs, the total cost of the continuous model is significantly lower than the sum of costs for models trained from scratch for every task.

### 4.3 DOMAIN ADAPTATION

All previous experiments had new classes in the 'new' data, often referred to as class-incremental learning in continual learning (De Lange et al., 2021). Here, we use the Adaptiope dataset (Ringwald & Stiefelhagen, 2021) to show that our method also works when new data contains no new classes, but the same classes from a different domain. In particular, the 'old' data consists of 123 categories of product images from a shopping website and the 'new' data are real-life images of the same products captured by users. The objective is to achieve strong performance across both domains, ideally with faster convergence than re-training the model from scratch on both datasets.

Figure 7 shows how our method outperforms both training from scratch and the naive continuous approach when including a new domain. The relative speed-up compared to the naive baselines is smaller than in the class-incremental case, but the final accuracy is considerably improved.

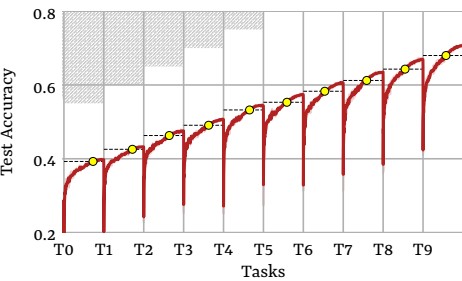 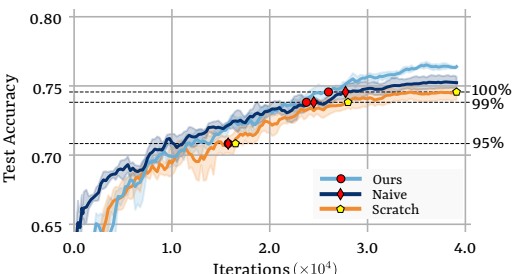

Figure 6: Test accuracy on the full CIFAR-100 when training a continuous model with our method on CIFAR100 $(50 + \sum_{i=1}^{10} 5)$. Yellow dots mark the $L_{100}$ iteration, where the continuous model outperforms the from-scratch model. The shaded area indicates maximum possible performance based on data availability (e.g., 55% during the first task $50 + 5$).

Figure 7: Domain adaptation results using the Adaptiope dataset. The 'old' data contains *product images*, the 'new' data has *real life* images. The test accuracy is on both domains. Although naively continuing to train is nearly as fast as our method in this setting, our final test accuracy is considerably higher.

Table 2: Results of our method across various datasets and splits. We report final test accuracy and relative speedup to reach similar performance compared to the scratch solution. The multiplier after the algorithm name indicates learning rate scheduling aggressiveness (e.g., Ours ($\times 0.5$) means the minimum learning rate is achieved at half the iterations of Ours ($\times 1$)). '/' indicates that the accuracy level was not reached. Table (a) presents results on different datasets, while Table (b) provides an in-depth analysis of CIFAR-100 with various class splits.

(a) More image classification benchmarks

| Dataset | Algorithm | Max Acc | $L_{99}$ | $L_{100}$ |
|---|---|---|---|---|
| CIFAR10 (8+2) | Scratch | 83.32 | $\times 1.55$ | $\times 1$ |
| | Continuous | 81.37 | / | / |
| | Ours ($\times 1$) | 83.95 | $\times 1.83$ | $\times 1.29$ |
| | Ours ($\times 0.5$) | **84.33** | $\times 3.61$ | $\times 2.92$ |
| Adaptiope-PI (100+23) | Scratch | 74.59 | $\times 1.14$ | $\times 1$ |
| | Continuous | 74.93 | $\times 1.17$ | 1.02 |
| | Ours ($\times 1$) | **77.51** | $\times 1.55$ | $\times 1.42$ |
| | Ours ($\times 0.5$) | 76.75 | $\times 2.36$ | $\times 2.24$ |
| ImageNet-100 (80+20) | Scratch | 62.34 | $\times 1.18$ | $\times 1$ |
| | Continuous | 62.21 | $\times 1.23$ | / |
| | Ours ($\times 1$) | **67.41** | $\times 2.16$ | $\times 1.97$ |
| | Ours ($\times 0.5$) | 66.26 | $\times 3.39$ | $\times 3.09$ |
| ImageNet-200 (180+20) | Scratch | 55.16 | $\times 1.18$ | $\times 1$ |
| | Continuous | 53.5 | / | / |
| | Ours ($\times 1$) | **59.62** | $\times 1.69$ | $\times 1.64$ |
| | Ours ($\times 0.5$) | 58.54 | $\times 2.84$ | $\times 2.75$ |

(b) Different ratios of old and new classes.

| Dataset | Algorithm | Max Acc | $L_{99}$ | $L_{100}$ |
|---|---|---|---|---|
| CIFAR100 | Scratch + L2-init | 67.90 | $\times 1.29$ | $\times 1.00$ |
| | Scratch ($\times 0.5$) | 67.14 | / | / |
| 90+10 | Ours ($\times 1.0$) | 69.42 | $\times 1.56$ | $\times 1.41$ |
| | Ours ($\times 0.5$) | 69.07 | $\times 2.95$ | $\times 2.65$ |
| | Ours ($\times 0.25$) | 68.42 | $\times 5.05$ | $\times 4.47$ |
| 70+30 | Ours ($\times 1.0$) | 69.47 | $\times 1.61$ | $\times 1.45$ |
| | Ours ($\times 0.5$) | 68.77 | $\times 2.84$ | $\times 2.56$ |
| | Ours ($\times 0.25$) | 68.26 | $\times 4.74$ | $\times 4.34$ |
| 50+50 | Ours ($\times 1.0$) | 69.09 | $\times 1.50$ | $\times 1.35$ |
| | Ours ($\times 0.5$) | 68.42 | $\times 2.70$ | $\times 2.37$ |
| | Ours ($\times 0.25$) | 67.57 | $\times 4.34$ | / |
| 30+70 | Ours ($\times 1.0$) | 68.81 | $\times 1.49$ | $\times 1.31$ |
| | Ours ($\times 0.5$) | 68.10 | $\times 2.56$ | $\times 2.30$ |
| | Ours ($\times 0.25$) | 66.93 | / | / |

## 4.4 OTHER DATASETS AND SCENARIOS

In Section 3, we demonstrated the effectiveness of each method using the CIFAR-100 (70+30) dataset for consistency across experiments. Table 2a extends these results to a range of datasets, including CIFAR-10, ImageNet-100, ImageNet-200, and the product images of Adaptiope. For each dataset, we compare the performance of models trained from scratch, a naive continuous model, and a continuous model using our full method. We report the relative speed-up to reach 99% and 100% of the scratch-trained model's accuracy, along with the final accuracy. Across all the different datasets we tested, our method consistently enhances the speed of convergence both for the $L_{99}$ and $L_{100}$ cases, often also surpassing the final accuracy of the scratch model.

Table 2b presents results for different class split ratios, including CIFAR-100 splits of (90+10), (70+30), (50+50), and (30+70). Our methods consistently accelerate training, with speed-up being more significant when the second data split is smaller. This aligns with intuition: when the ini-

tial 'old' data is larger, the old model already has more knowledge, requiring fewer updates when exposed to new data, compared to training from scratch, which treats both splits equally.

## 5 CONCLUSION

Throughout this paper, we have shown that the computational cost that is paid to train models on old data should not be treated as something that is lost, but rather as a starting point for further training whenever new data becomes available. We studied the four main components of the traditional SGD update rule – initialization, objective function, data selection and hyperparameters – and showed that each of them individually can contribute to faster convergence on old and new data combined when starting from an old model. These aspects are thoroughly tested to be robust and effective across a wide range of scenarios. Our proposals should be seen as starting points to work further towards solutions that are even more effective in re-using old models, and saving computational costs across the board in machine learning development and applications.

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

# A APPENDIX

## A.1 COST ESTIMATION

To train a machine learning model there is both a memory (storage) and compute cost. Here we will work out an example for training a Vision Transformer (ViT) (Vaswani, 2017) on ImageNet 21k. This dataset is about 1.13 terabytes. Cloud storage for common providers averages around 0.023$ per GB each month, which would be around 30.13$ per month (Google, 2024a). However, local storage is much cheaper than cloud storage. Today hard disk storage is about 11$ per TB (McCallum, 2023), which can last for 10 years or more.

ViT models were trained on 8 Nividia P100 gpus for 3.5 days (Vaswani, 2017), which cost $1.46 per hour on Google Cloud, which would be $981 in total for the entire training (Google, 2024b). Buying the GPUs would be much more expensive, with prices for a single A100 GPU above $10.000 (Nvidia, 2024). Modern A100 GPUs are about 3.5 times faster on real-life work loads (Rio, 2023), but more expensive to rent. At 4$4.05 per hour, the total price would still be $777.5.

## A.2 HYPERPARAMETERS

Unless specified otherwise, all experiments are trained with the Adam optimizer (Kingma, 2014), with default settings of $\beta_1 = 0.9$, $\beta_2 = 0.999$ and no weight decay. The starting learning rate is equal to $0.001$ and the batch size is consistently $128$ in all experiments. Unless specified differently, a cosine scheduler is used where the minimal learning rate $1e-6$ is reached after $39.100$ iterations, which is equal to $50$ epochs using the complete CIFAR100 dataset. All experiments use cropping and random horizontal flip augmentations.

## A.3 INITIALIZATION

Figure 8 shows different values of $\alpha$ in the shrink and perturb update rule. In genreal, shrinking more gives better results, although it slows down learning results at the start of learning. Shrinking enough is necessary and not doing so may lead to suboptimal accuracies because of loss of plasticity. In Figure 9 different values of $\beta$ are tested, which add various levels of noise to the initialization. There is no visible effect of the size of this parameter, which is roughly in line with (Ash & Adams, 2020).

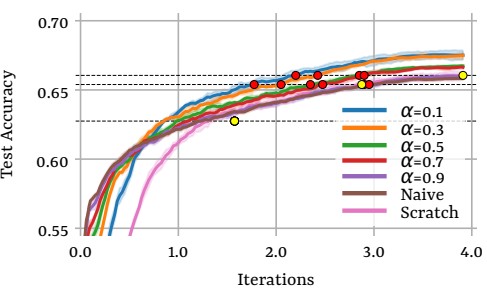 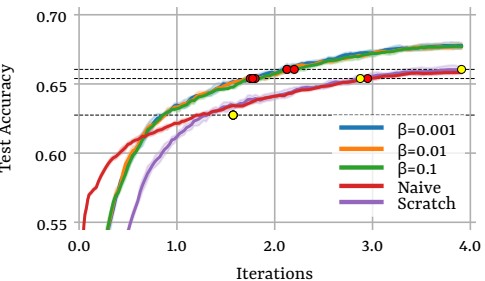

Figure 8: Grid test of the $\alpha$ (shrink) hyperparameter in the shrink and perturb update rule.

Figure 9: Grid test of the $\beta$ (perturb) hyperparameter in the shrink and perturb update rule

## A.4 REGULARIZATION

Figure 10 shows the influence of the $\lambda$ parameter in the $L2$-init regularization, which controls the strength of the regularization. A too low value will not have any effect, while setting this value too high, will lead to slower than necessary learning speeds.

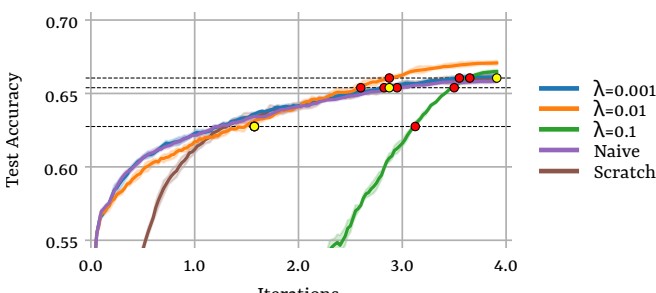

Figure 10: Ablation of the $\lambda$ parameter in the $L2$-init regularization.

## A.5 BATCH COMPOSITION

Figure 11 shows how fast examples are learned during the first task of CIFAR100 (70+30). The x-axis shows the epochs, while each row indicates a single sample. The examples on the bottom are green almost from the start of training, these are the ones that are very easy: the model almost has no difficulty learning them, which also means they do not carry a lot of information. The ones on the top are almost completely red: they are never learned. These examples are equally not very useful: they are so hard the model can not learn them anyway (which may be a result of bad labeling, bad images etc.).

In Figure 12, we ablate two of the parameters of the 'easy / hard' sampling process. $c$ indicates what proportion of the easy and hard examples is influenced. e.g. $c = 0.2$ means that 20% procent of the examples is affected, or the 10% easiest and the 10% hardest. $r$ indicates the probability that the easy and hard examples are sampled in a batch, relative to the examples in the middle. e.g. with $r = 0.1$, an easy or hard example is 10 times less likely to be in a minibatch. The result with $r = 0$ shows that we can even get rid of these examples, reducing the memory requirements for this method. When $r = 0$, this becomes the method proposed by (Hacohen & Tuytelaars, 2024), on which this idea is based.

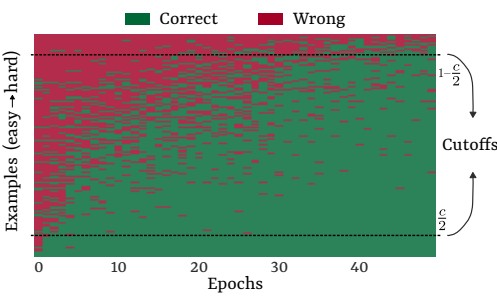 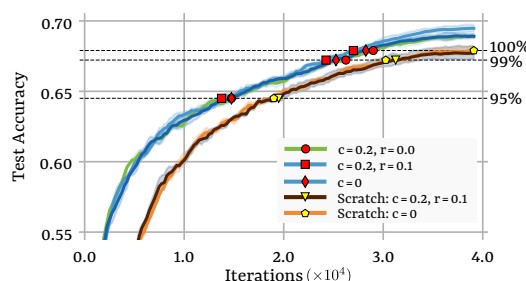

Figure 11: How fast examples are learned during the initial learning phase of training on the old data of CIFAR100 70+30 (i.e. the first 70 classes). Some examples are almost learned instantly (bottom), others never (top). The cutoffs indicate both the 10% easiest and hardest examples.

Figure 12: Test accuracy of CIFAR100 on the $70 + 30$ benchmark. Removing the easiest and hardest examples ($r = 0$) barely has an influence. Sampling them with a low probability ($r = 0.1$) allows to learn a little faster on the continuous model, without such an effect on the from scratch model.

## A.6 SCHEDULERS

Figure 13 shows an experiment on CIFAR100 (70+30), with a multistep learning rate scheduler (which reduces the learning rate by a fixed factor at fixed steps) rather than a cosine learning rate scheduler. This scheduler reaches the same conclusion, although in this case the iterations where the scheduler kicks in has a much larger influence than in the cosine scheduler case. In the continuous case, the learning rate is kept too high for too long, preventing the model to learn the last details, while the from scratch model is still learning.

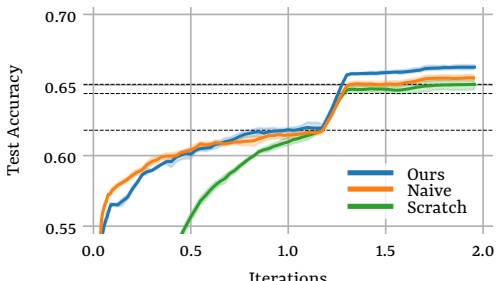

Figure 13: When using multi-step schedulers the qualitative result stays the same, although cosine schedulers are better suited to this task of continuous learning.

## A.7 GRADIENT NORMS

Figure 14 shows the average gradient norm of examples of old and new data in the CIFAR100 (70+30) baseline during training of the naive baseline. While at the very start the gradient of new data is considerably higher, this difference has completely disappeared after more than 100 iterations, which is nearly immediately considering $40.000$ iterations in total. This explains mostly why it is not useful to oversample new data, which would indicate that new data is more important to learn than remembering the old data, which is not the goal.

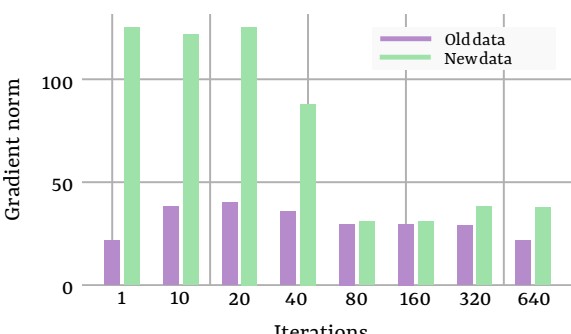

Figure 14: Gradient norms of 'old' and 'new' data during training of the naive baseline on CI-FAR100 (70+30).

## A.8 LOSS PLATEAU

Figure 15 shows how the loss of the naive solution plateaus earlier than when training from scratch, thus allowing to schedule learning rates earlier. However, it is not because such a loss plateau is reached that the final result will be better, it merely indicates that the results won't get any better when training longer than from the point the plateau is reached. In fact, as can be seen in Figure 15, the loss starts to increase again as overfitting takes place. In this sense, it might even be necessary to schedule early enough to obtain the best possible results.

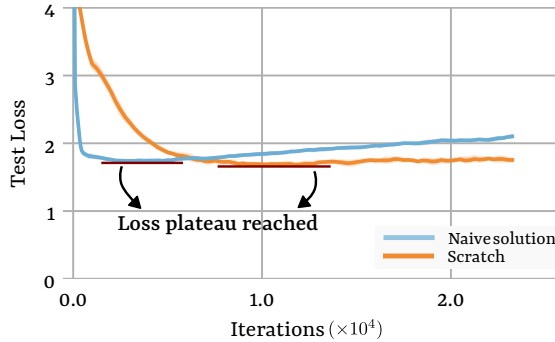

Figure 15: The continuous naive baseline loss plateaus earlier than that of the from scratch baseline.

