# OpenReview forum: "Same Accuracy, Twice As Fast: Continual Learning Surpasses Retraining From Scratch"
_ICLR.cc/2025/Conference — Submitted to ICLR 2025_

### Official Review · Reviewer_wjkS · 2024-10-16

**Soundness:** 2
**Presentation:** 2
**Contribution:** 1
**Rating:** 3
**Confidence:** 3

**Summary:**

Overall, this paper first defines a new setting of "continual learning", where the old data, the current data, and the model that has been trained on the old data are all accessiable. After that, it presents a bag of tricks that can enable the model to achieve much faster convergence in this new setting of "continual learning".

**Strengths:**

Overall, I believe that the problem that this paper aims to handle is an important problem. In other words, when we already have a model that has been trained over part of the dataset, it is a waste to simply re-train everything from scratch.

Also, the paper present this problem in a well-motivated way.

**Weaknesses:**

(See questions)

**Questions:**

Overall, I believe that this paper is not ready to be accepted. Below are my concerns.

1. The main concern I have w.r.t. this paper is over its defination that its setting is a ``continual learning'' setting. Specifically, correct me if I am wrong, yet, based on my understanding of continual learning, the catastrophic forgetting is always a key problem in this area, no matter over which of its sub-settings such as class-incremental learning, domain-incremental learning, and so on. Yet, with the access of the old data, the paper seems to define a continual learning setting that is essentially different from the general continual learning concept. This worries me that whether this paper fits itself in a wrong area for the research community. For example, if my understanding is not wrong, warm starting is a method that is kind of just solving the problem that this paper aims to solve. Yet, no one would recognize warm starting as a continual learning method.

2. Furthermore, the paper highlights that it makes several "first-step methods". Yet, I am confused over the first step here. If I am not wrong, the paper seems to be highly inspired by previous methods and thus seems to be closely related to them. For example, the initialization part seems to be closely related to Ash & Adams (2020), the batch composition part seems to be closely related to Hacohen & Tuytelaars (2024), and so on. Specifically, while I can admit that adapt some method to a new area is a contribution, as mentioned in my previous concern, I doubt whether the authors do propose a "new" setting, not to say that the contribution behind adaptation can be small.

3. My last concern lies in the generalizability of the proposed method. Specifically, it seems to me that the method just try some approaches and observe that they work on a few (small) datasets. I am not sure whether this can well justify and demonstrate the effectiveness of the proposed method.

Given my above concerns, I believe that this paper is not ready to be accepted in its current version.

---

> ### Author Response · Authors · 2024-11-22
>
> Thank you for taking your time to review our paper. We believe we have slightly different definitions of what is continual learning, which we clarify below. Thank you for acknowledging that this is indeed an important problem to study.
>
> **About the definition of continual learning.** Definitions are a hard thing to agree on. We interpret continual learning as learning from any kind of data that is not i.i.d. Often, research additionally imposes a memory constraint, which then leads to catastrophic forgetting when data is not i.i.d. When defining continual learning as learning from non-i.i.d. data, this setting is perfectly interpretable as continual learning. However,  if it is necessary to avoid confusion with other works, it is possible to change continual learning to ‘continuous’ learning in our paper, with extra clarification on the (dis)similarities with continual learning in the main text.
>
> **On the “first step methods”.** All the works that inspired study different subproblems. We are the first to connect them together and show that in this scenario, together they form a practically usable algorithm. It is e.g. not self explanatory that the result of Ash and Adams translate to this setting, as they only test benchmarks where both old and new data are from the same distribution.
>
> **Generalizability.** We have tested many datasets and experiments in Tables 2a and 2b, Figure 6 and 7 and additional hyperparameters settings in the Appendix. None of them showed different results. It is always possible to include another dataset, yet we found the selection we have made relevant to test various aspects of our proposed method.

---

> > ### Comment · Reviewer_wjkS · 2024-11-23
> >
> > After reading the authors' rebuttal. I decide to keep my current score. This is due to two reasons.
> >
> > 1. The authors seem to also agree that they are not continual learning but just their so-called "continuous" learning. Considering this, I believe that, even if this paper is worthy to these non-continual but "continuous" learning people, the authors need to make tremendous changes to their paper. This makes this submission better to deserve another round of review, instead of being accepted just this time.
> >
> > 2. Meanwhile, it seems to me that the authors only incompletely make their rebuttal. For example, they incompletely answer the generalizability question in my rebuttal. Meanwhile, during their rebuttal to Reviewer wpKw, when they justify that this setting is also confirmed by other papers to be relevant, they leave a TODO there at the place that I guess is supposed to be the citation of those other papers. Thus, I think the authors by themselves also cannot clearly support their relevance between their proposed setting and the community.
> >
> > Considering the above, I believe that the submission is not ready to be accepted in this round of review, and at least needs another round of submission.

---

> ### Author Response · Authors · 2024-11-23
>
> Thank you for noticing the cut off sentence and the TODO in one of the other comments, that was an oversight from our side. The references are added in the appropriate place now and the sentence is completed.

---

### Official Review · Reviewer_2cbZ · 2024-10-25

**Soundness:** 3
**Presentation:** 3
**Contribution:** 2
**Rating:** 5
**Confidence:** 4

**Summary:**

This paper explores the problem of efficient model training through the lens of continuous training. It focuses on four aspects: initialization, regularization, data selection and hyper-parameter tuning. For each aspect, it introduces specific techniques to enhance the training efficiency. The experiments are performed on several image classification datasets using a ResNet-18 network. By integrating all the proposed methods, the paper demonstrates improved performance while reducing training costs.

**Strengths:**

1. It is interesting to explore efficient training from the perspective of continuous training.
2. This paper provides an initial investigation using different training techniques and shows their effectivenss. The strategies employed perform well even in multi-step training scenarios.
3. This paper is easy to follow.

**Weaknesses:**

1. This paper only demonstrates its feasibility on small-scale models and datasets. However, efficiency gains are more critical when dealing with large models and datasets. Please also provide the validation on larger model and datasets. It would be better to see results on ImageNet-1k with ViTs and comparing with training from scratch or other pre-train based continual leanring baselines such as [A].
2. The use of the term "continual learning" may be misleading, as it conficts with its wildly accepted meaning in the field. It would be better to use other phrase like "continuous training" instead. Additionally, providing a clear definition of the task is necessary to avoid confusion.
3. There are some discrepancies in the reported results. For instance, in Table 2(b), the cost reduction ratios are ×1.61 and ×1.45 for CIFAR-100 (70+30) with full data, and ×4.74 and ×4.34 with 25\% data. However, in Table 1, the reported values are ×1.96, ×1.69, and ×5.73, ×5.32, respectively. These inconsistencies should be addressed and clarified.

[A] SLCA: Slow Learner with Classifier Alignment for Continual Learning on a Pre-trained Model. Zhang et al. ICCV 2023.

**Questions:**

Line 315, "More formally, they define learning speed ls of a sample $x_j$ as the epoch $e$ in which sample is classified correctly" is unclear. $e$ is not used in Equation 6.

---

> ### Author Response · Authors · 2024-11-22
>
> Thank you for reviewing our paper and taking the time to give feedback. Your comments have helped to improve the paper further. Below, we hope to address some of the additional concerns you had.
>
> **CL-methods.** Methods like SLCA are interesting and strong methods when memory is the core limitation, but they do not reach from scratch accuracy and require additional computation compared to from-scratch learning, see e.g. Prabhu et al. [a], who showed that many traditional continual learning methods cannot compete with simple rehearsal when considering the computational cost.
>
> **Is it continual learning?** Continual learning is about learning from data that is not i.i.d. distributed. It is true that often a memory restriction is imposed, but a part of this work wants to argue that settings with no restrictions on memory but restrictions on computational complexity may be interesting too. However, if it is necessary to avoid confusion with other works, it is possible to change the term continual learning to ‘continuous’ learning, with extra clarification on the (dis)similarities in the main text.
>
> **Different results in Table 1 and 2.** You are correct about the inconsistencies. The results are correct, but it is a little confusing, so thanks for catching this. They are different because in Table 2(b), the methods are compared with scratch+L2-init, rather than bare scratch in Table 1. Scratch+L2-init gave approximately the best results ($L_{100}$) for the scratch model, so we chose to compare to that baseline in Table 2b. We will update the text to make this clearer. Note that ‘25%’ refers to the length of the training, i.e. the cosine scheduler is reduced to ¼-th of the length of the from-scratch scheduler.
>
> [a] Prabhu, A., Cai, Z., Dokania, P., Torr, P., Koltun, V., & Sener, O. (2023). Online continual learning without the storage constraint. arXiv preprint arXiv:2305.09253.

---

> > ### Comment · Reviewer_2cbZ · 2024-11-27
> >
> > I have check the rebuttal and and the comments from fellow reviewers. The concerns regarding the generalizability of the method (Weakness 1) and the misleading definition of the problem setting (Weakness 2) were not adequately addressed in the rebuttal. These issues were also raised by other reviewers. Consequently, I have decided to slightly lower my rating.

---

### Official Review · Reviewer_5bTG · 2024-11-01

**Soundness:** 3
**Presentation:** 3
**Contribution:** 2
**Rating:** 3
**Confidence:** 3

**Summary:**

This paper tackles the practical aspect of continual learning setup, wherein access to both new and old data exists and the major cost of training newer models is on the computation side. With the set of previously known techniques, authors show that when used,  in combination they provide significant speed-up in the training of a new model under the described continual learning paradigm. The experiments are done on the standard datasets which shows the validity of the claims.

**Strengths:**

1. This method shows the combination of model initialization, learning rate scheduling, batch composition and the loss function, affect the rate at which the model learns on new data.
2. Though the method is simple and a combination of known techniques, it provides good improvement on some of the datasets.
3. The paper is well-written and easy to read.

**Weaknesses:**

1. The major issue with this work is how generalizable are the claims on the varied set of backbone architecture and datasets. For eg. ViT backbones, downstream tasks, like object detection or segmentation, or varied datasets like ImageNet or wider domain shifts like in WiLDS[1]. It will make the claims stronger with more extensive results. As section 4.3 suggests the claim might not hold under domain shifts.
2. Will the method benefit already existing CL methods(ones mentioned in Sec1.2)? if the claims are general, it should benefit the CL methods and one can hope for the best of both, performance-boost and computation efficiency.
3. The comparisons with existing CL methods are missing in the experiments in terms of performance and compute efficiency. $L_r$ can be computed for them too. This makes the assessment of the contribution difficult.
4. The test accuracy is reported for combined new and old data. It will be better to see the accuracy on them separately.


[1]:https://arxiv.org/pdf/2211.14238

**Questions:**

See the weakness section.

---

> ### Author Response · Authors · 2024-11-22
>
> Thanks you for your time and your comments. We hope that below, we can clarify some of the comments you made and the issues you encountered.
>
> **More benchmarks.** We have tested our method on many different benchmarks and domain shifts, as is explicitly tested in Section 4.3 and Figure 7. Tables 2a and 2b contain results on many different datasets and benchmarks. The proposed problems and architectures would further broaden our claims, but given the many experiments already included in the paper, we did not find it feasible to include even more experiments (there are always more experiments we can think of).
>
> **Continual learning methods.** It is true that other CL methods could be included, but they never reach the same accuracy as a model that is trained from scratch, even when they have full memory. Furthermore, many continual learning methods do add computational complexity to combat forgetting. If they do not reach the same accuracy, their $L_{100}$ would be undefined, but if they do, they would likely require a higher computational cost than we do. We are the first to propose a baseline in this setting, so it is hard to directly compare to other methods.
>
>
> **New and old data.** We will include results on new and old data in the appendix. It does give insights, but since in the end a combined result on both old and new data is the goal, we reported the combined performance.

---

> > ### Comment · Reviewer_5bTG · 2024-11-24
> >
> > Thank you for your rebuttal.
> > The justification for the lack of extensive benchmarking seems weak. Since there is no theoretical backing for the methods, the only way to know the robustness of the claims is with empirical results on varied backbones and tasks.
> >
> > I'd keep my rating the same and encourage authors to include experiments to strengthen their paper.

---

### Official Review · Reviewer_vBZf · 2024-11-03

**Soundness:** 3
**Presentation:** 3
**Contribution:** 2
**Rating:** 6
**Confidence:** 3

**Summary:**

This paper considers a new continual learning setting where the old data and new data are both accessible for training and the efficiency is the main concern. For each task, the straightforward baselines include training the model from scratch with both old and new data and fine-tuning the model trained on old data with new data. The former serves as a strong baseline considering the accuracy but is computationally expensive. The latter suffers from a much worse accuracy. Thus, this paper investigates a bag of tricks used in SGD training to improve both the accuracy and efficiency of the way that fine-tunes the model pretrained on old data. Empirical study verifies the effectiveness of proposed method.

**Strengths:**

- The paper is generally well-written and easy to follow.

- The empirical results are good.

- It verifies a bag of tricks which may help the practitioners to continually train a model with satisfactory accuracy and efficiency.

**Weaknesses:**

- Some details are not clear. For example, in Fig. 3, the curve corresponding to the legend "L2" means "scratch + L2" or "naive + L2"? If it is the later, how about the curve of "scratch + L2"?

  I don't quite understand the descriptions from Line 415-419. I am confused about how the authors compute different costs. Besides, for
  Fig. 6, I don't understand what the curve between $T_{i-1}$ and $T_i$ means. And I am not clear how the authors draw the curve.

- There is a typo in Eq. (6), the $x$ and $y$ on the right side of equal sign should be $x_j$ and $y_j$.

- For the batch composition trick, it can be shared across the fine-tuning way and the scratch way. From the empirical results shown in Table 1, applying this trick to the "continuous" and the "scratch" yields the same speed considering $L_{100}$ and almost the same speed considering $L_{99}$. Also, by comparing "init+reg" and "init+reg+data" (continuous), it seems that the batch composition trick does not work. To me, the results are reasonable. I don't understand why batch composition can yield larger gain when combined with continuous model compared to that combined with scratch model. Do the authors have any explanations for this?

**Questions:**

- The authors applied a bag of tricks which may be proposed in previous works to the considered continual learning setting. Can the authors provide any insights why those tricks work in the continual learning setting? Are there any other tricks which may also benefit the accuracy and efficiency of continual learning?

---

> ### Author Response · Authors · 2024-11-22
>
> We would like to thank you for your time and comments. They helped use thinking about the problem a little deeper. Thanks for appreciating our work.
>
> **L2+naive.** L2 in Figure 3 is L2+naive. For scratch training with either L2-init or L2 did not make a large difference. To not overload the figure we chose to not include L2+scratch in the same figure. We will add it to the appendix and comment on this in the main paper.
> Clarification of repeated task experiment. Figure 6 is in essence a repetition of 10 figures like Fig. 2-5. Each line between $T_{i-1}$ and $T_{i}$ shows the training on all old and new data available in that task. The only difference is that the evaluation is on all classes of CIFAR100, where the future ones evidently have zero accuracy. To improve the clarity, we will add an explanation to the figure caption.
>
> **Typo.** Thanks for catching the typo in Equation 6, this will be updated.
>
> **Why batch composition may work.** The idea of the batch composition is that the easiest data does not need to be repeated as often as the other data, as the continuous data already knows this data well, in contrast to the from scratch model, which has never seen this easy data. The hardest data was never learned by this model, so it is likely not going to contribute a lot of information when it is rehearsed. The from scratch model has not learned anything yet, so it may benefit more from seeing diverse (easy and hard) samples than the continuous model does. Concerning the speed-up there is indeed only a small difference when init+reg is used, but it does allow a higher final accuracy than without it.
>
> **Insights.** The reason we propose these four ‘tricks’ is because they act on the four main components of any optimizer. With them, we show that every aspect of the optimization can provide potential improvements. We approach this problem from the optimization perspective, because fundamentally, only the initialization is different in the continual case, which largely impacts the optimization trajectory. There may be more and different ways to continue with this problem, but this way, we sought to provide a principled way of tackling this problem.

---

### Official Review · Reviewer_wpKw · 2024-11-04

**Soundness:** 2
**Presentation:** 2
**Contribution:** 2
**Rating:** 3
**Confidence:** 4

**Summary:**

The work "proposes" a continual learning framework/analysis based on optimization to reduce the computational cost during the continual learning of models.

**Strengths:**

The authors conduct a thorough analysis of the SGD optimization scheme on a continual learning (CL) setup and open new insights into a new direction. The analysis is limited and not thorough. The literature on CL has moved way ahead, considering the core data and model assumptions that this work has considered. However, such an analysis is also important.

**Weaknesses:**

The core contributions of this work are completely minimal and rely on the analysis of standard deep-learning optimization schemes, especially with regard to the parameter update rule, in a continual learning (CL) scheme. The results are not dense and the experimental setup is very basic. Rigorous experiments are to be done on a wide scale. The authors claim that "old" and "new" data are usually available, which is very impractical by today's standards. Please take a look at recent works [1, 2, 3]. However, many works have proposed CL methods with no access to past data + (data and parametric)-efficient ways. This work doesn't meet the standards of an ICLR submission, however. The analysis is decent, but the experiments are not thorough with no clear "proposed method". My suggestion would be to conduct such analysis on a larger scale, covering a wide range of cases - training setups (CIL, TIL), model backbones, challenging scenarios, etc.



[1] Wang, Z., Zhang, Z., Lee, C.Y., Zhang, H., Sun, R., Ren, X., Su, G., Perot, V., Dy, J., Pfister, T.: Learning to prompt for continual learning. In: CVPR. pp. 139–149

[2] James Seale Smith, Leonid Karlinsky, Vyshnavi Gutta, Paola Cascante-Bonilla, Donghyun Kim, Assaf Arbelle, Rameswar Panda, Rogerio Feris, and Zsolt Kira. Coda-prompt: Continual decomposed attention-based prompting for rehearsal-free
continual learning. In CVPR, pages 11909–11919, 2023.

[3] Dahuin Jung, Dongyoon Han, Jihwan Bang, and Hwanjun Song. Generating instance-level prompts for rehearsal-free continual learning. In ICCV, pages 11847–11857, 2023.

**Questions:**

1. It is not clear to me how important saving "computational time" is. Could the authors benchmark against FLOPs?
2. If the "old" data isn't available, can the authors benchmark their work against prompt-based CL methods like L2P [1]? Or can a similar analysis be done with a ViT backbone?
3. Why do the authors use a 70+30 split on CIFAR-100? How about a uniform split in tasks?
4. How would the analysis hold with rapid domain shifts?
5. Is there a typo in Fig 6 caption/analysis? In the main text (4.2), a CIFAR-100 (70+30) was used.
6. How is the analysis with an Adam/AdamW optimizer? Is this analysis model invariant and holds everywhere?



[1] Wang, Z., Zhang, Z., Lee, C.Y., Zhang, H., Sun, R., Ren, X., Su, G., Perot, V., Dy, J., Pfister, T.: Learning to prompt for continual learning. In: CVPR. pp. 139–149

---

> ### Author Response · Authors · 2024-11-22
>
> We would like to thank the reviewer for their time and feedback. We hope that below, we can clarify some of the questions you had.
>
> **About the setting**. The idea of this paper is to specifically study a scenario where all data is available. This may not be a common setting, but as we explain in the introduction, and confirmed by other papers ([a-c]), it is a very relevant setting. Especially in industry applications, data is often stored and available. Parameter efficient finetuning is a useful approach without old data available, but it does not allow to reach the same accuracy as a model that is trained from scratch on both old and new data. Even outside a continual learning setting, PEFT methods do not allow to reach high accuracy when pretraining and downstream data are evaluated as one large dataset. It would be an interesting future direction to incorporate ideas of those algorithms further into our work, but there first needs to be a basis that allows continuously trained models to reach the same accuracy as a model that is trained from scratch, which we provide.
>
>
> **FLOPs vs. `computational time’.** We use computational time measured by the number of iterations, as it is a good proxy to measure e.g. FLOPs or MACS as long as batch sizes and model architectures are consistent across compared algorithms. The number of iterations is directly proportional to FLOPs, hence reducing iterations will reduce the FLOPs.
>
>
> **What if no old data is available?** We specifically study continual learning where all data is available. There are excellent works (e.g. the ones that you cite) that study scenarios where there is no (or little) old data available. Our method is not intended to be used in such scenarios, similar to how prompt based methods would not be used when all old and new data is available.
>
>
> Results with a **balanced task split** are in Table 2b and reach comparable results.
>
> **Rapid domain shifts.** Figure 7 explicitly studies domain shifts and Figure 6 includes many small tasks.
>
> **Different benchmark in Figure 6?**. Figure 6 explicitly studies the case where many tasks are added over time, hence it cannot use the same 70+30 benchmark. To add many tasks, the first task is reduced to 50 classes here, such that 10 tasks of 5 classes can be added.
>
> **Adam.** We tried to test as many hyperparameters as possible, many of which are reported in the ablations. Experiments are mostly performed with Adam, see the Appendix. We did not find configurations where the results were significantly different than the ones reported.
>
> [a] Prabhu, A., Al Kader Hammoud, H. A., Dokania, P. K., Torr, P. H., Lim, S. N., Ghanem, B., & Bibi, A. (2023). Computationally budgeted continual learning: What does matter?. In Proceedings of the IEEE/CVF Conference on Computer Vision and Pattern Recognition (pp. 3698-3707).
>
> [b] Verwimp, E., Aljundi, R., Ben-David, S., Bethge, M., Cossu, A., Gepperth, A., ... & van de Ven, G. M. Continual Learning: Applications and the Road Forward. Transactions on Machine Learning Research.
>
> [c] Shi, H., Xu, Z., Wang, H., Qin, W., Wang, W., Wang, Y., ... & Wang, H. (2024). Continual learning of large language models: A comprehensive survey. arXiv preprint arXiv:2404.16789. (Section 6.3)

---

> > ### Comment · Reviewer_wpKw · 2024-11-25
> >
> > I'd like to thank the authors and fellow reviewers for actively participating in this discussion. Based on the rebuttal provided by the authors, I'll continue with the same rating as given earlier. The major reason is that such analysis based on a standard optimization scheme in a continual learning setup seems trivial. All of us researchers perform such standard tweaks, in our daily experiments,  while optimizing our model - be it in a CL setup or in a setup with domain shifts (like continual test-time domain adaptation).
> >
> > In addition, my question based on "rapid domain shifts" meant each task comes from a different domain (which is extremely challenging) - and seems unanswered. I also second the reviews provided by Reviewer 5bTG. The claims in the paper seem to be very casually accepted and $\textit{might not}$ hold for denser architectures and tougher CL setups.

---

### Meta-Review · Area_Chair_Z6Nf · 2024-12-16

**Metareview:**

This paper proposes a new setup for continual learning where both old and new data, as well as a pre-trained model, are accessible. It demonstrates that combining existing techniques significantly accelerates training in this setup, validated through experiments on standard datasets. The reviewers highlights several major concerns about the practicality and generalisability of the proposed approach. Firstly, the assumption that both old and new data are typically available is considered unrealistic in many real-world scenarios, where retaining access to old data is often restricted due to privacy, storage, or regulatory constraints. Secondly, doubts are raised about the generalisability of the claims, as the experiments lack sufficient evidence to demonstrate the effectiveness of the proposed methods across a diverse range of backbone architectures and datasets. Lastly, the use of the term "continual learning" is flagged as potentially misleading, as it diverges from the widely accepted definition in the field, which typically refers to scenarios without access to old data.

**Additional Comments On Reviewer Discussion:**

Five expert reviewers highlight several concerns. While some concerns have been addressed in the rebuttal, several major ones remain unresolved. As noted by reviewers wpKw, 5bTG, 2cbZ and wjkS, the generalisability of the proposed method is unclear. More precisely, As noted by Reviewer 5bTG, the justification for the lack of extensive benchmarking is weak. This is mainly because there is no theoretical backing for the proposed method, and thus the only way to support the claims is with empirical results on larger datasets and varied backbones and tasks. In addition, as noted by reviewers wjkS and 2cbZ, the use of the term "continual learning" is misleading, as it diverges from the widely accepted definition in the field (i.e., scenarios without access to old data). In summary, three reviewers recommended "reject", one rated the paper "marginally below the acceptance threshold," and one reviewer, while recommending "marginally above the acceptance threshold," did not strongly advocate for acceptance.,

---

### Decision · Program_Chairs · 2025-01-22

Reject